# Rooting ability of rice seedlings increases with higher soluble sugar content from exposure to light

Wanlai Zhou[1,2,3], Zhiyong Qi[1], Jing Chen[3], Zhijian Tan[2], Hongying Wang[2], Chaoyun Wang[2]*, Zhenxie Yi[3]*

1 Institute of Urban Agriculture, Chinese Academy of Agricultural Sciences, Chengdu, Sichuan, China,
2 Institute of Bast Fiber Crops, Chinese Academy of Agricultural Sciences, Changsha, Hunan, China,
3 College of Agronomy, Hunan Agricultural University, Changsha, Hunan, China

☯ These authors contributed equally to this work.
* ibfcwcy@139.com (CW); yizhenxie@126.com (ZY)

**Data Availability Statement:** All relevant data are within the manuscript.

## Abstract

Rooting ability of rice seedling for mechanical transplanting has a large impact on grain yield. This study explored the relationship between endogenous soluble sugar content and rooting ability of rice seedlings. We placed 15-day-old rice seedlings in controlled environment cabinets with stable light and sampled after 0, 3, 6, 9, 12, and 24 hours of light to measure their soluble sugar content, nitrate content, starch content, soluble protein content and rooting ability. The soluble sugar content of the rice seedlings before rooting increased rapidly from 65.1 mg g$^{-1}$ to 126.3 mg g$^{-1}$ in the first 9 hours of light and then tended to stabilize; however, few significant changes in the other physiological indices were detected. With the light exposure time increasing from 3 hours to 12 hours, the rooting ability measured with fresh weight, dry weight, total length, and number of new roots increased by 91.7%, 120.0%, 60.6% and 30.3%, respectively. Rooting ability was related more closely to soluble sugar content than to nitrate-nitrogen content of rice seedlings before rooting and their correlation coefficients were 0.8582–0.8684 and 0.7045–0.7882, respectively. The stepwise regression analysis revealed that the soluble sugar content before rooting explained 73.6%–75.4% of the variance, and the nitrate-nitrogen content explained an additional 7.3%–14.2% of the variance in rooting ability, indicating that compared with nitrate-nitrogen content, soluble sugar content of rice seedlings before rooting was more dominant in affecting rooting ability. This study provides direct evidence of the relationship between the rooting ability and endogenous soluble sugar content of rice seedlings.

## Introduction

With an increasingly scarce rural labor force, mechanical transplanting has become an important cultivation method to replace hand transplanting in Chinese rice production [1]. In mechanical transplanting practices, rice seedlings usually undergo "transplantation shock" in which phase the growth of transplanted seedlings becomes stagnant for several days but invests

**Funding:** This research and the APC was funded by the Young Scientists Fund of the National Natural Science Foundation of China, grant number 31701372 (http://www.nsfc.gov.cn/) and the Natural Science Foundation of Hunan Province, grant number 2018JJ3583 (http://61.187.87.55/egrantweb/#). The funders had no role in study design, data collection and analysis, decision to publish, or preparation of the manuscript.

**Competing interests:** The authors have declared that no competing interests exist.

its energy to establish a new root system. Only when new roots grow, new tillers begin to emerge [2]. Therefore, the initiation and growth of new roots directly affect the emergence of new tillers of mechanically transplanted rice in the field and has a large impact on grain yield. Rooting ability is usually used to assess the ability of rice seedlings to form and grow new roots after transplantation [3, 4]. Specifically, rooting ability is measured with the length, number, and/or quality of the new roots of root-cut rice seedlings cultured in water for 3–10 days [5]. Numerous field experiments have indicated that a strong rooting ability of rice seedlings was positively correlated with root vigor, drought resistance, effective panicle formation, and even the rice grain yield [6–9]. Thus, in recent years, there has been considerable effort put into studying the factors affecting the rooting ability of mechanically transplanted rice and their control measures [10–14].

In many field experiments, the rooting ability of rice seedlings raised by different seedling-raising techniques was quite different [15, 16]. For example, the rooting ability of dry-raised seedlings was significantly higher than that of wet-raised seedlings [17], and the rooting ability of seedlings raised in seedling trays with matting of bast fiber seedling film was significantly higher than that without the bast fiber seedling film [18]. In these studies, the increase in rooting ability was usually accompanied by an increase in soluble sugar content of the rice seedlings [17, 18], suggesting that soluble sugars may play an important role in rice rooting activity.

Some experiments based on sugar-containing medium were done to explore the relationship between plant root development and sugar [19–22]. Fumio Takahashi et al. [23] reported that sucrose, glucose, and fructose greatly stimulated the induction of adventitious roots, but mannose or sorbitol did not. However, when high concentrations of sugar were added to the growth medium, the induction of adventitious roots was suppressed. In addition, numerous other studies have confirmed that nitrate-nitrogen in the growth medium stimulated lateral root formation and increased root length [24, 25]. Some studies showed that the effect of sugar on plant root development was related to the ratio of sugar to nitrogen in the growth medium [26]. These research on the regulation of root development by the sugar-controlled medium essentially studied the effects of exogenous sugars in the root zone (sugars from the growth medium) on plant root development, which may be exactly the same as that of the sugars in the plant (endogenous sugar), also may not be exactly the same or even completely different. As the sugar content of living plants is hard to control, an understanding of the relationship between endogenous sugars and plant root development is still relatively inadequate.

The soluble sugar content of most plants fluctuates daily under natural light. In most cases, the soluble sugar content of plant under light increases first and then decreases. For example, the sucrose content in maize leaves reached its peak after 4 hours of light [27], in tomato seedlings leaves reached its peak after 5–7 hours of light [28], and in soybean leaves reached its peak after 6–8 hours of light [27, 29]. However, Zhou et al. [30] showed that the soluble sugar content in lettuce leaves and petioles continued to increase under 72 hours of light from fluorescent lamps. The content of fructose in tomato leaves increased continuously under natural light, while sucrose and glucose increased first and then decreased [31], indicating that the fluctuation pattern of soluble sugar content in plant under light varies among different sugars.

In this study, we obtained rice seedlings different in soluble sugar content but roughly uniform in other aspects by controlling the illumination time, and then conducted contrast tests of their rooting ability, aimed to explore the relationship between endogenous sugars and rooting ability of rice seedlings, so as to provide guidance for finding out some measures that can be directly used to increase the rooting ability of rice seedlings for mechanical transplanting.

## Materials and methods

### Rice seedling raising

Rice seedlings used for the rooting ability test were raised in an experimental field of the Institute of Bast Fiber Crops, Chinese Academy of Agricultural Sciences (N28˚12' E112˚44') in Changsha, Hunan, China. The rice variety was Xiangwanxian No. 17. Plastic seedling trays (58 cm × 28 cm × 2.5 cm deep) were used for raising the rice seedlings. Both the rice variety and the seedling tray have been widely used in mechanically transplanted rice planting in southern China. Pre-germinated seeds were evenly sown in the seedling soil on June 10, 2018 at a sowing rate of 120 g per tray. After sowing, the seedling trays were placed on the pre-leveled seedling bed in the experimental field, and received only natural light throughout the seedling stage.

### Treatments and their implementation

According to the difference in duration time of light to rice seedlings before rooting test, six treatments were set up in the experiment, which were 0, 3, 6, 9, 12, 24 hours of light before rooting. According to our conjecture, the soluble sugar content of rice seedlings under different treatment would be quit different. Each tray of seedlings was used as a repeat, and there were three replicates. The specific implementation is as follows:

On the evening of June 25, 2018 (around 6 pm), three trays of uniform rice seedlings were selected from the seedlings cultivated in the field and placed in three controlled environment cabinets (MGC-400H, Shanghai Yiheng Science Instrument Co., Ltd.) set to dark, at a temperature of 25˚C and a relative humidity of 70%. After about 14 hours of darkness (i.e., at 8 am the next day), we turned on the lights in the controlled environment cabinets and set the light intensity at 20000 Lux, the temperature at 28˚C, and the relative humidity at 70%. Rice seedlings were taken out after 0 (the initial point of light), 3, 6, 9, 12, and 24 hours of light. At each time, about 100 rice seedlings were selected from each tray for rooting ability test and physiological analysis.

### Rooting ability test and physiological analysis

Rice seedlings were carefully washed to remove the soil and divided into 3 groups: one group was immediately used for the rooting ability test, one group was kept at 4˚C to measure the plant height, chlorophyll, carotenoid, and soluble protein content of the rice seedlings before rooting, and the third group was first dried at 105˚C for 0.5h and then dried to a constant weight at 80˚C to determine the soluble sugar, starch, and nitrate-nitrogen content of the rice seedlings before rooting.

The rooting ability was measured as follows: all the roots of the rice seedlings were carefully cut off at the root base and then the root-cut rice seedlings were planted in cultivation boards floating on tap water in a box, then the box was placed in a controlled environment cabinet (MGC-400H, Shanghai Yiheng Science Instrument Co., Ltd) set at a temperature of 28˚C, and a relative humidity of 70%. After 72 hours, the seedlings were taken out and the surface water on the new roots was blot-dried with paper. The seedlings were then placed on a scanner, the morphology of new roots was scanned, and we used Root nav v1.8.1 [32] to analyze the number and total length of the new roots. 24 rice seedlings from each sample were used to test rooting ability and all the new roots of the 24 seedlings were then carefully cut off, and their fresh weight was determined, then they were dried to a constant weight at 105˚C to determine their dry weight. Lastly, the root-cut rice seedlings were dried at 105˚C for 0.5h and then dried to a constant weight at 80˚C to determine the soluble sugar, starch, and nitrate-nitrogen contents

of rice seedlings after rooting. We expressed the number, total length, fresh weight, and dry weight of new roots with the average values of the 24 seedlings measured.

The soluble sugar and nitrate-nitrogen contents were determined by the phenol and salicylic acid methods according to Li [33]. The starch content was measured on the rice plant samples after the extraction of soluble sugars by heating them in a perchloric acid environment to hydrolyze the starch into soluble sugars and then determining the soluble sugar content; the starch content was determined by multiplying the soluble sugar content by 0.9 [33]. Leaf chlorophyll and carotenoid was extracted with 95% ethanol and determined by colorimetry [33]. The soluble protein content was determined by the Coomassie Brilliant Blue G-250 method [33].

### Statistical analysis

The data were analyzed using SAS 8.2. A Tukey's HSD test after One-Way ANOVA was used to compare the differences in plant height, chlorophyll, soluble protein, soluble sugar, nitrate-nitrogen, and starch content among rice seedlings after exposure to different durations of light. A t-test was used to compare the differences in soluble sugar, nitrate-nitrogen, and starch content between the rice seedlings before and after rooting. We used a Pearson correlation coefficient and partial correlation coefficient to evaluate the correlation between seedling physiological indices (plant height, chlorophyll, soluble protein content, soluble sugar content, nitrate-nitrogen content, and starch content) and rooting ability indices (fresh weight, dry weight, total length, and number). We used a stepwise regression analysis to analyze each independent variable potentially affecting rooting ability.

## Results

### Plant height, leaf chlorophyll, carotenoid, and soluble protein content of rice seedlings after different durations of light

As shown in Fig 1, the plant height, leaf chlorophyll, carotenoid and soluble protein content of rice seedlings after different durations (0, 3, 6, 9, 12 and 24 hours) of light did not show statistically significant difference, indicating that these physiological indexes of rice seedlings kept basically the same under a relatively short period (less than 24 hours) of light, and it was just as we expected.

### Soluble sugar, starch, and nitrate-nitrogen content of rice seedlings after different durations of light

The results from the rice seedlings before rooting showed that the soluble sugar content of the rice seedlings increased significantly with light (Table 1). The soluble sugar content increased rapidly from 65.1 mg g$^{-1}$ to 126.3 mg g$^{-1}$ (i.e., an increase of 94%) in just the first 9 hours and then tended to stabilize, eventually reaching 130.9 mg g$^{-1}$ at the end of the 24 hours of light exposure. The nitrate-nitrogen content was relatively stable under light exposure. The starch content of the rice seedlings fluctuated during the 24 hours of light exposure, however, the differences in nitrate-nitrogen and starch content among different times was statistically insignificant.

The contents of soluble sugar, starch, and nitrate-nitrogen of the rice seedlings after rooting decreased significantly (Table 1), implying a large amount of sugar catabolism in rooting process, which may provide energy and structure matters for plant physiological activities. Specifically, the average soluble sugar content declined from 104.6 mg g$^{-1}$ to 83.7 mg g$^{-1}$, or a decrease of 20.1%; the average starch content declined from 232.5 mg g$^{-1}$ to 215.2 mg g$^{-1}$, or a

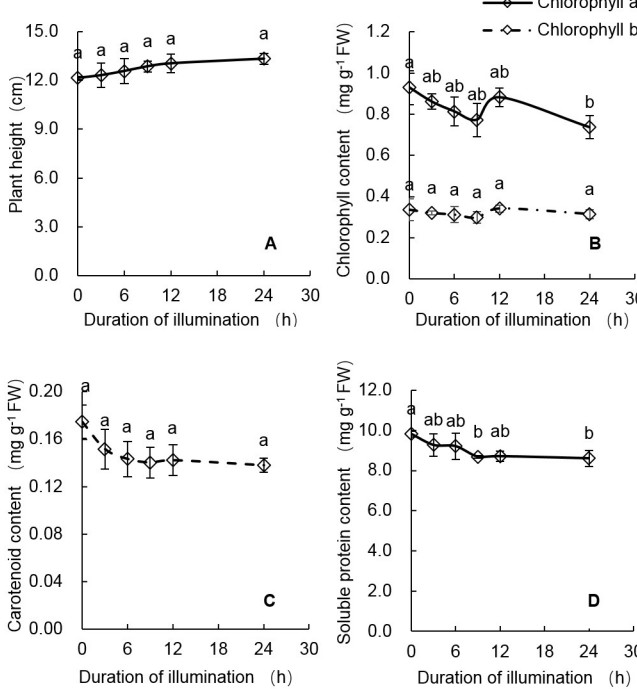

**Fig 1. Plant height (A), leaf chlorophyll (B), carotenoid (C), and soluble protein content (D) of rice seedlings after 3, 6, 9, 12, and 24 hours of light exposure.** Error bars represent SE (n = 3). The same letter in the same series represent no significant difference according to Tukey's HSD test at the 0.05 level.

**Table 1. Soluble sugar, starch, and nitrate-nitrogen contents before and after rooting in rice seedlings after 0, 3, 6, 9, 12, and 24 hours of light exposure.**

| Test time | Duration of light | Soluble sugar content | Starch content | Nitrate-nitrogen content |
|---|---|---|---|---|
| | h | mg g$^{-1}$ DW | mg g$^{-1}$ DW | mg g$^{-1}$ DW |
| Before rooting | 0 (initial) | 65.1 ± 4.2c | 237.8 ± 24.4a | 3.3 ± 0.2a |
| | 3 | 81.4 ± 6.8c | 241.2 ± 13.8a | 3.3 ± 0.2a |
| | 6 | 98.8 ± 1.1b | 231.4 ± 11.6a | 3.4 ± 0.1a |
| | 9 | 126.3 ± 5.6a | 244.2 ± 9.7a | 3.4 ± 0.1a |
| | 12 | 125.5 ± 6.1a | 232.5 ± 15.4a | 3.8 ± 0.3a |
| | 24 | 130.9 ± 8.9a | 207.6 ± 11.6a | 3.8 ± 0.2a |
| After rooting | 0 (initial) | 77.6 ± 2.5ab | 205.9 ± 14.4a | 2.9 ± 0.5a |
| | 3 | 73.5 ± 8.2b | 235.0 ± 6.9a | 2.5 ± 0.1ab |
| | 6 | 90.8 ± 3.1a | 198.1 ± 8.4a | 2.3 ± 0.3ab |
| | 9 | 91.2 ± 7.9a | 232.3 ± 27.6a | 2.6 ± 0.1ab |
| | 12 | 78.3 ± 4.0ab | 211.3 ± 28.3a | 2.7 ± 0.3ab |
| | 24 | 90.6 ± 4.5a | 208.8 ± 4.3a | 2.0 ± 0.2b |
| T-test | t | 3.22 | 2.84 | 8.77 |
| | p | 0.0041 | 0.0076 | < 0.0001 |

Values are means ± standard error (n = 3). Means with the same letter in the same column at the same test time are not significantly different according to Tukey's HSD test at the 0.05 level.

decrease of 7.4%; and the average nitrate-nitrogen content declined from 3.5 mg g$^{-1}$ to 2.5 mg$^{-1}$, or a decrease of 27.9%.

## Rooting ability of rice seedlings after different durations of light

The rooting ability of rice seedlings were measured with the fresh weight, dry weight, total length, and number of new roots, all of which showed basically the same trends with light duration (Fig 2); this was confirmed by the high correlations among fresh weight, dry weight, total length, and number of new roots (Table 2). In the first 3 hours of light, the rooting ability of rice seedlings decreased slightly, but the difference was not statistically significant. With the light exposure time increasing from 3 hours to 12 hours, the rooting ability of the rice seedlings increased rapidly (Fig 2). Specifically, the fresh weight of new roots increased from 6.0 mg to 11.5 mg, or an increase of 91.7% (Fig 2A); the dry weight of new roots increased from 0.5 mg to 1.1 mg, or an increase of 120.0% (Fig 2B); the total length of new roots increased from 46.2 mm to 74.2 mm, or an increase of 60.6% (Fig 2C); and the number of new roots increased from 3.3 to 4.3, or an increase of 30.3% (Fig 2D). Unlike the previous rapid increases, when light exposure time was greater than 12 hours, the increase in rooting ability measured with the fresh weight and dry weight of new roots slowed down (Fig 2A and 2B), while the rooting ability measured with total length and number of new roots tended to be constant (Fig 2C and 2D).

For rice seedlings before rooting, it can be seen from the Pearson's correlations in Table 2 that rooting ability (fresh weight, dry weight, total length and number of new roots) had positive linear correlations with plant height, soluble sugar and nitrate-nitrogen content, while negative linear correlations with soluble protein and starch content. We also found significant linear relationships of soluble sugar content with plant height, soluble protein and nitrate-nitrogen content ($p<0.05$), and of nitrate-nitrogen with soluble protein and starch content

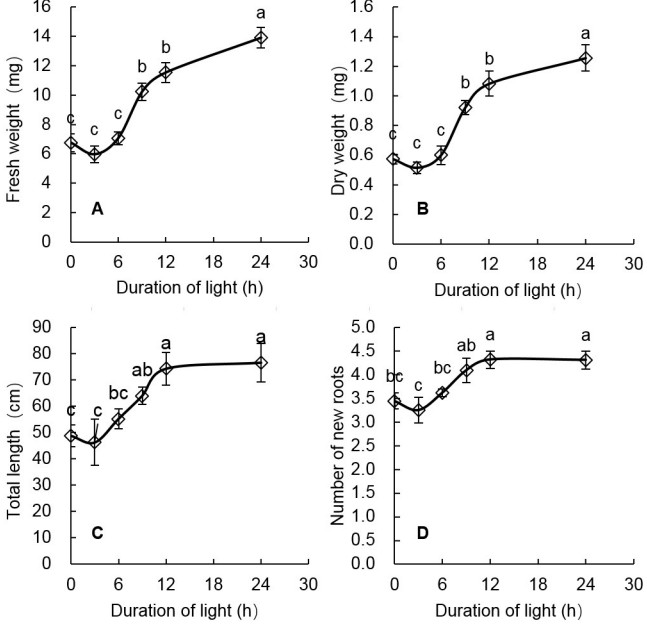

**Fig 2. Rooting ability as measured with the fresh weight (A), dry weight (B), total length (C) and number of new roots (D) of rice seedlings after 0, 3, 6, 9, 12, and 24 hours of light.** Error bars represent SE (n = 3). The same letter in the same series represent no significant difference according to Tukey's HSD test at the 0.05 level.

**Table 2. Pearson's correlations (n = 18) of rice seedling physiological indices with rooting ability indices.**

| Test time | Index | Fresh weight | Dry weight | Total length | Number |
|---|---|---|---|---|---|
| Before rooting | Height (cm) | r = 0.6633 p = 0.0027 | r = 0.6310 p = 0.0050 | r = 0.5205 p = 0.0268 | r = 0.6315 p = 0.0049 |
| | Chlorophyll-a content | r = -0.4867 p = 0.0476 | r = -0.4321 p = 0.0833 | r = -0.4725 p = 0.0555 | r = -0.3761 p = 0.1368 |
| | Chlorophyll-b content | r = -0.0649 p = 0.8113 | r = -0.0032 p = 0.9905 | r = -0.0549 p = 0.8401 | r = 0.0027 p = 0.9920 |
| | Carotenoid content | r = -0.3370 p = 0.2018 | r = -0.3226 p = 0.2230 | r = -0.4009 p = 0.1238 | r = -0.3170 p = 0.2316 |
| | Soluble protein content | r = -0.6809 p = 0.0019 | r = -0.7347 p = 0.0005 | r = -0.6910 p = 0.0015 | r = -0.6709 p = 0.0023 |
| | Soluble sugar content | r = 0.8665 p < 0.0001 | r = 0.8684 p < 0.0001 | r = 0.8582 p < 0.0001 | r = 0.8646 p < 0.0001 |
| | Starch content | r = -0.4739 p = 0.0469 | r = -0.4581 p = 0.0559 | r = -0.4913 p = 0.0384 | r = -0.4755 p = 0.0461 |
| | Nitrate content | r = 0.7075 p = 0.0010 | r = 0.7045 p = 0.0011 | r = 0.7882 p = 0.0001 | r = 0.7814 p = 0.0001 |
| After rooting | Soluble sugar content | r = 0.3472 p = 0.1581 | r = 0.3404 p = 0.1669 | r = 0.2417 p = 0.3338 | r = 0.2849 p = 0.2519 |
| | Starch content | r = 0.0142 p = 0.9568 | r = 0.0486 p = 0.8531 | r = 0.0367 p = 0.8888 | r = 0.0119 p = 0.9637 |
| | Nitrate content | r = -0.3505 p = 0.1539 | r = -0.3459 p = 0.1597 | r = -0.2010 p = 0.4238 | r = -0.0891 p = 0.7252 |
| Fresh weight | | | r = 0.9887 p < 0.0001 | r = 0.9319 p < 0.0001 | r = 0.9138 p < 0.0001 |
| Dry weight | | | | r = 0.9220 p < 0.0001 | r = 0.8971 p < 0.0001 |
| Total length | | | | | r = 0.9307 p < 0.0001 |

($p<0.05$). In view of this, a partial correlation analysis was conducted. Both the partial-correlations between rooting ability and plant height, soluble protein with soluble sugar content held constant and the partial-correlations between rooting ability and starch content with nitrate-nitrogen content held constant were not significant ($p>0.05$), indicating that there were no substantial correlations between plant height, soluble protein, starch content and rooting ability. This partial correlation analysis indicates that only soluble sugar content and nitrate-nitrogen content had correlations with rooting ability, and they were independent of each other. However, rooting ability was related more closely to soluble sugar content than to nitrate-nitrogen content; and their correlation coefficients were 0.8582–0.8684 and 0.7045–0.7882, respectively. For rice seedlings after rooting, however, we found no significant relationships of the contents of soluble sugar, nitrate-nitrogen, and starch with the rooting ability.

We conducted a multiple regression with the rice seedling physiological indices that had significant relationships with rooting ability (i.e., plant height, soluble protein content, soluble sugar content, starch content, and nitrate-nitrogen content of the rice seedlings before rooting) as independent variables with stepwise regression analysis (Table 3). No matter which rooting ability index was used as the dependent variable, the soluble sugar content was introduced in the first step, followed by the nitrate-nitrogen content, and then the stepwise regression process was terminated. In the overall multiple regression model, the soluble sugar content explained 73.6%–75.4% of the variance ($R^2 = 0.7364$–$0.7541$), while the nitrate-nitrogen content explained an additional 7.3%–14.2% of the variance in rooting ability ($R^2 = 0.0727$–

**Table 3. Stepwise regression analysis of factors related to rooting ability.**

| Dependent variable | Selection summary | | | | | Parameter estimates | | | Variance analysis of regression model |
|---|---|---|---|---|---|---|---|---|---|
| | Step | Variable entered | Partial $R^2$ | Model $R^2$ | F, *p* | β | t, *p* | VIF | F, *p* |
| **Fresh weight** | 1 | SBR | 0.7507 | 0.7507 | F = 48.19 $p < 0.0001$ | 0.68452 | t = 5.30 $p < 0.0001$ | 1.4392 | F = 35.63 $p < 0.0001$ |
| | 2 | NBR | 0.0754 | 0.8261 | F = 6.50 $p = 0.0222$ | 0.32933 | t = 2.55 $p = 0.0222$ | 1.4392 | |
| **Dry weight** | 1 | SBR | 0.7541 | 0.7541 | F = 49.08 $p < 0.0001$ | 0.68972 | t = 5.35 $p < 0.0001$ | 1.4392 | F = 35.81 $p < 0.0001$ |
| | 2 | NBR | 0.0727 | 0.8268 | F = 6.30 $p = 0.0240$ | 0.32347 | t = 2.51 $p = 0.0240$ | 1.4392 | |
| **Total length** | 1 | SBR | 0.7364 | 0.7364 | F = 44.70 $p < 0.0001$ | 0.60841 | t = 5.63 $p < 0.0001$ | 1.4392 | F = 54.20 $p < 0.0001$ |
| | 2 | NBR | 0.1420 | 0.8784 | F = 17.52 $p = 0.0008$ | 0.45209 | t = 4.19 $p = 0.0008$ | 1.4392 | |
| **Number** | 1 | SBR | 0.7476 | 0.7476 | F = 47.39 $p < 0.0001$ | 0.62314 | t = 5.82 $p < 0.0001$ | 1.4392 | F = 55.21 $p < 0.0001$ |
| | 2 | NBR | 0.1328 | 0.8804 | F = 16.66 $p = 0.0010$ | 0.43718 | t = 4.08 $p = 0.0010$ | 1.4392 | |

SBR, soluble sugar content before rooting. NBR, nitrate-nitrogen content before rooting. β, standardized partial regression coefficient. VIF, variance inflation. All variables left in the model were significant at α = 0.05, and no other variable met the 0.05 significance level for entry into the model.

0.1420), further indicating that the soluble sugar content before rooting had the closest relationship with rooting ability.

## Discussion

In this study, the soluble sugar content of rice seedlings after different durations of time was significantly different, while other physiological indexes were basically the same, so such rice seedlings can be used as experimental material to study the relationship between endogenous soluble sugar content and rooting ability of rice seedlings.

Our results showed that the soluble sugar content of rice seedlings under short-term durations light increased rapidly, which is in full agreement with previous observations [27–30]. It is well known that the main end products of photosynthesis in green plants are sucrose and/or starch [34, 35]. For rice, it's normally sucrose [36, 37], therefore, the rapid increase in soluble sugar content in rice seedlings under light should be attributed to the accumulation of photosynthates. The soluble sugar content of rice seedlings tended to be stable (around 125 mg g$^{-1}$ DW) with the continuous illumination, while the starch content did not increase at this time, suggesting that with the accumulation of soluble sugar in rice seedlings, a feedback regulation that can inhibit photosynthesis probably occurred [38–40], thus slowing down or even preventing the accumulation of photosynthates.

There was a sharp decrease of soluble sugar content during rooting process in this study, suggesting that the soluble sugars in rice seedlings underwent catabolism during the rooting process, which can provide energy and structural substances for root growth and development. The higher the soluble sugar content is, the more materials and energy can be provided. Therefore, it's natural that there was a strong positive correlation between rooting ability and soluble sugar content and that the soluble sugar plays an important role in determining the rooting ability of rice seedlings.

However, the rooting process of plant is affected by various substances that affect the initiation, elongation and thickening of roots, respectively. For example, increase in carbohydrate

content of rice plants increased their rooting rate of differentiated root primordia [41], and nitrate-nitrogen can promote formation and elongation of lateral root [24, 25]. Therefore, the rooting ability reflects the comprehensive results under the influence of different substances. In our experiment, the soluble sugar content of rice seedlings increased in the early stage of light (0–3 hours), but the rooting ability did not increase or even decreased in the same period, suggesting that besides soluble sugar, the rooting ability of rice seedlings should also be related to some other substances which may restrict the rooting activity of rice seedlings. From 3 hours to 9 hours, both the soluble sugar content and rooting ability of rice seedlings increased linearly with time, indicating that the soluble sugar from photosynthesis at this stage may be the main factor affecting the root development of rice seedlings. From 12 hours to 24 hours, the soluble sugar content of rice seedlings didn't increase significantly, neither did the number and total length of new roots, while the weight of new roots increased significantly, showing that when the light duration exceeds 12 hours, increasing the time of light could not promote the initiation and elongation new roots, however, it can promote the thickening of new roots. All of these demonstrated that the substances determining rooting ability in rice seedlings under light changed with the duration of light. However, the types of these substances and their relationship with rooting ability still need to be further studied.

In addition, there was also a significant strong correlation between the nitrate-nitrogen content and the rooting ability of rice seedlings, indicating that the nitrate-nitrogen might also be involved in determining the rooting ability of rice seedlings, which is consistent with the previous results [17]. However, correlation analysis in this study showed that rooting ability had closer correlation with soluble sugar content than with nitrate-nitrogen content, and the multiple regression analysis further showed that the soluble sugar content before rooting explained more than 70% of the variance in rooting ability while the nitrate-nitrogen content explained much less, indicating that compared with nitrate-nitrogen content, soluble sugar content was more dominant in determining rooting ability. It agreed with the conclusion from the study about root development based on sugar-containing medium that sugars were one of the most important factors regulating the initiation and growth of plant roots [42]. In some field experiments, the nitrate-nitrogen content of rice seedlings decreased, while their soluble sugar content as well as rooting ability increased [18]; in addition, researchers obtained completely opposite conclusions about the relationship between rooting ability and nitrogen content of rice seedlings, while their conclusion on the positive correlation between rooting ability and soluble sugar content was consistent [41, 43]. These results demonstrate that the positive correlation of rooting ability with soluble sugar content is more common than with nitrate-nitrogen content, which also supports the conclusion of this study.

## Conclusion

This study provides direct evidence of the relationship between the rooting ability and endogenous soluble sugar content of rice seedlings. Although both soluble sugar and nitrate-nitrogen content of rice seedlings before rooting had strong positive correlations with rooting ability, the soluble sugar content was more dominant in affecting rooting ability.

Given that the soluble sugar content of the rice seedlings increased linearly with light time when it was less than 12 hours, we can infer that the rooting ability of rice seedlings would be greatly enhanced by transplanting them after receiving enough light (more than 3 hours). We recommended transplanting rice seedlings in the afternoon or evening, or using artificial light regulation when raising the seedlings. These practices would accelerate rejuvenation and tillering of rice seedlings after transplanting and consequently ensure high grain yields.

## Author Contributions

**Data curation:** Zhiyong Qi.

**Investigation:** Jing Chen, Hongying Wang.

**Methodology:** Zhenxie Yi.

**Project administration:** Wanlai Zhou.

**Supervision:** Chaoyun Wang.

**Writing – original draft:** Wanlai Zhou, Zhiyong Qi.

**Writing – review & editing:** Zhijian Tan.

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
