## [Decision Letter · Decision Letter 0]

10 Sep 2020

PONE-D-20-24008

Rooting ability of rice seedlings increases with higher soluble sugar content from exposure to light

PLOS ONE

Dear Dr. Zhou,

Thank you for submitting your manuscript to PLOS ONE. After careful consideration, we feel that it has merit but does not fully meet PLOS ONE’s publication criteria as it currently stands. Therefore, we invite you to submit a revised version of the manuscript that addresses the points raised during the review process.

We look forward to receiving your revised manuscript.

Kind regards,

Saddam Hussain

Academic Editor

PLOS ONE

Journal Requirements:

Reviewers' comments:

Reviewer's Responses to Questions

**Comments to the Author**

1. Is the manuscript technically sound, and do the data support the conclusions?

Reviewer #1: Yes

Reviewer #2: Yes

Reviewer #3: Yes

2. Has the statistical analysis been performed appropriately and rigorously? 

Reviewer #1: Yes

Reviewer #2: Yes

Reviewer #3: Yes

3. Have the authors made all data underlying the findings in their manuscript fully available?

Reviewer #1: Yes

Reviewer #2: Yes

Reviewer #3: Yes

4. Is the manuscript presented in an intelligible fashion and written in standard English?

Reviewer #1: Yes

Reviewer #2: Yes

Reviewer #3: Yes

5. Review Comments to the Author

Reviewer #1: Study submitted by Zhou et al on "Rooting ability of rice seedlings increases with higher soluble sugar content from exposure to light" explains correlation between endogenous sugars with rooting ability under different light durations. In its present form, I feel authors fails to highlight the novelty of this study. During active growth phase most of the energy coming through respiration is utilised for growth process for example rooting in the early seedling establishment phase. This energy is certainly coming by the breakdown of simple sugars. Thus, it is very obvious to get a strong and positive correlation between rooting ability and endogenous sugars. Moreover, more light would increase amount of photo-assimilates until a saturation point. On the other hand, rooting ability is a complex trait which is tightly regulated by genetic background and genotype x environment interaction. Authors should discuss in more details, how increase in assimilates partitioned towards root. As roots could be main sink during early growth, traits such as leaf area, photosynthetic ability and root architecture may be crucial to determine overall rooting ability.

Reviewer #2: The research experiment was well executed with proper statistical support. The results were well presented and discussed with proper references. The comments of earlier reviewers were properly addressed and the same reflected in the MS. With this the MS entitled "Rooting ability of rice seedlings increases with higher soluble sugar content from exposure to light" can accepted for publication.

Reviewer #3: Overall this is a well written manuscript. However a minor improvement is required in the introduction section. Exposure to light is the main treatment of this manuscript but in the last paragraphs of introduction section only 1-2 general sentences have been added about the variation of soluble sugars in response to light exposure. I suggest add more specific literature in last paragraphs of the introduction section to justify the of selection of this treatment.

6. PLOS authors have the option to publish the peer review history of their article (what does this mean?). If published, this will include your full peer review and any attached files.

Reviewer #1: No

Reviewer #2: **Yes: **Seetharam Kaliyamoothy

Reviewer #3: No

---

## [Author Response · Author response to Decision Letter 0]

22 Sep 2020

Response to Reviewer 1 Comments

Point 1: In its present form, I feel authors fails to highlight the novelty of this study. During active growth phase most of the energy coming through respiration is utilized for growth process for example rooting in the early seedling establishment phase. This energy is certainly coming by the breakdown of simple sugars. Thus, it is very obvious to get a strong and positive correlation between rooting ability and endogenous sugars. Moreover, more light would increase amount of photo-assimilates until a saturation point. On the other hand, rooting ability is a complex trait which is tightly regulated by genetic background and genotype x environment interaction. Authors should discuss in more details, how increase in assimilates partitioned towards root. As roots could be main sink during early growth, traits such as leaf area, photosynthetic ability and root architecture may be crucial to determine overall rooting ability

Response 1: Thank you very much for your professional and kind suggestions. We agree with your assessment, your suggestions have opened our mind and we have reorganized the discussion section of our article to highlight the novelty of our study. Based on the experimental data we got, we adjusted our discussion points to “the substances determining rooting ability in rice seedlings under light changed with the duration of light” and “soluble sugar content was more dominant in determining rooting ability compared with nitrate-nitrogen content”. Indeed, it is very obvious to get a strong and positive correlation between rooting ability and endogenous sugars. However, our results showed that it’s not so simple. Rooting ability is a complex trait, and our article demonstrated the influence of light on rooting ability and the complexity of the effect of light duration on rooting ability, and compared the relative importance of the effects of endogenous soluble sugar and nitrate nitrogen. Based on this, we put forward some suggestions that can guide the rice mechanical transplanting of rice. 

Response to Reviewer 2 Comments

Point 1: The research experiment was well executed with proper statistical support. The results were well presented and discussed with proper references. The comments of earlier reviewers were properly addressed and the same reflected in the MS. With this the MS entitled "Rooting ability of rice seedlings increases with higher soluble sugar content from exposure to light" can accepted for publication.

Response 1: Thank you for your comments. 

Response to Reviewer 3 Comments

Point 1: Overall this is a well written manuscript. However a minor improvement is required in the introduction section. Exposure to light is the main treatment of this manuscript but in the last paragraphs of introduction section only 1-2 general sentences have been added about the variation of soluble sugars in response to light exposure. I suggest add more specific literature in last paragraphs of the introduction section to justify the of selection of this treatment.

Response 1: Thank you very much for your professional and kind suggestions. We have made corresponding modifications according to your suggestions. In the introduction, we use a separate paragraph (line 74-83) to introduce the fluctuation of soluble sugar content in plants under light with detailed data.

---

## [Decision Letter · Decision Letter 1]

8 Oct 2020

Rooting ability of rice seedlings increases with higher soluble sugar content from exposure to light

PONE-D-20-24008R1

Dear Dr. Zhou,

We’re pleased to inform you that your manuscript has been judged scientifically suitable for publication and will be formally accepted for publication once it meets all outstanding technical requirements.

Kind regards,

Saddam Hussain

Academic Editor

PLOS ONE

Additional Editor Comments (optional):

Reviewers' comments:

Reviewer's Responses to Questions

**Comments to the Author**

1. If the authors have adequately addressed your comments raised in a previous round of review and you feel that this manuscript is now acceptable for publication, you may indicate that here to bypass the “Comments to the Author” section, enter your conflict of interest statement in the “Confidential to Editor” section, and submit your "Accept" recommendation.

Reviewer #1: All comments have been addressed

Reviewer #3: All comments have been addressed

2. Is the manuscript technically sound, and do the data support the conclusions?

Reviewer #1: Yes

Reviewer #3: Yes

3. Has the statistical analysis been performed appropriately and rigorously? 

Reviewer #1: Yes

Reviewer #3: Yes

4. Have the authors made all data underlying the findings in their manuscript fully available?

Reviewer #1: Yes

Reviewer #3: Yes

5. Is the manuscript presented in an intelligible fashion and written in standard English?

Reviewer #1: Yes

Reviewer #3: Yes

6. Review Comments to the Author

Reviewer #1: (No Response)

Reviewer #3: (No Response)

7. PLOS authors have the option to publish the peer review history of their article (what does this mean?). If published, this will include your full peer review and any attached files.

Reviewer #1: **Yes: **RAJEEV NAYAN BAHUGUNA

Reviewer #3: No

---

## [Editor Report · Acceptance letter]

12 Oct 2020

PONE-D-20-24008R1 

Rooting ability of rice seedlings increases with higher soluble sugar content from exposure to light 

Dear Dr. Zhou:

I'm pleased to inform you that your manuscript has been deemed suitable for publication in PLOS ONE. Congratulations! Your manuscript is now with our production department. 

Kind regards, 

on behalf of

Dr. Saddam Hussain 

Academic Editor

PLOS ONE